# The genome and phenome of the green alga *Chloroidium sp.* UTEX 3007 reveal adaptive traits for desert acclimatization

David R Nelson[1,2]*, Basel Khraiwesh[1,2], Weiqi Fu[1], Saleh Alseekh[3],
Ashish Jaiswal[1], Amphun Chaiboonchoe[1], Khaled M Hazzouri[2],
Matthew J O'Connor[4], Glenn L Butterfoss[2], Nizar Drou[2], Jillian D Rowe[2],
Jamil Harb[3,5], Alisdair R Fernie[3], Kristin C Gunsalus[2,6], Kourosh Salehi-Ashtiani[1,2]*

[1]Laboratory of Algal, Synthetic, and Systems Biology, Division of Science and Math, New York University Abu Dhabi, Abu Dhabi, United Arab Emirates; [2]Center for Genomics and Systems Biology, New York University Abu Dhabi, Abu Dhabi, United Arab Emirates; [3]Max Planck Institute of Molecular Plant Physiology, Potsdam, Germany; [4]Core Technology Platform, New York University Abu Dhabi, Abu Dhabi, United Arab Emirates; [5]Department of Biology and Biochemistry, Birzeit University, Birzeit, Palestine; [6]Center for Genomics and Systems Biology and Department of Biology, New York University, New York, United States

**Abstract** To investigate the phenomic and genomic traits that allow green algae to survive in deserts, we characterized a ubiquitous species, *Chloroidium sp. UTEX 3007*, which we isolated from multiple locations in the United Arab Emirates (UAE). Metabolomic analyses of *Chloroidium sp. UTEX 3007* indicated that the alga accumulates a broad range of carbon sources, including several desiccation tolerance-promoting sugars and unusually large stores of palmitate. Growth assays revealed capacities to grow in salinities from zero to 60 g/L and to grow heterotrophically on >40 distinct carbon sources. Assembly and annotation of genomic reads yielded a 52.5 Mbp genome with 8153 functionally annotated genes. Comparison with other sequenced green algae revealed unique protein families involved in osmotic stress tolerance and saccharide metabolism that support phenomic studies. Our results reveal the robust and flexible biology utilized by a green alga to successfully inhabit a desert coastline.

*For correspondence: drn2@nyu.edu (DRN); ksa3@nyu.edu (KS-A)

**Competing interests:** The authors declare that no competing interests exist.

## Introduction

Green algae play important ecological roles as primary biomass producers and are emerging as viable sources of commercial compounds in the food, fuel, and pharmaceutical industries. However, relatively few species of green algae have been characterized in depth at the genomic and metabolomic levels (*Koussa et al., 2014*; *Salehi-Ashtiani et al., 2015*; *Chaiboonchoe et al., 2016*). Also, recent genomics studies lack accompanying phenotype studies that could provide valuable context (*Bochner, 2009*; *Chaiboonchoe et al., 2014*). Analyses of this scope are necessary to better understand the ecology and physiology of microscopic algae (microphytes), both at the local and global scales, and to optimize their cultivation and yield of bioproducts for industrial applications (*Abdrabu et al., 2016*). Currently, species with superior growth characteristics for large-scale cultivation remain understudied, under-developed, and under-exploited (*Fu et al., 2016*).

Our study focuses on the green alga – formerly identified as *Chloroidium* sp. DN1 and accessioned at the Culture Collection of Algae at the University of Texas at Austin (UTEX) as *Chloroidium sp. UTEX 3007*, which we recently isolated in a screen for lipid-producing algae (*Sharma et al.,*

**eLife digest** Single-celled green algae, also known as green microalgae, play an important role for the world's ecosystems, in part, because they can harness energy from sunlight to produce carbon-rich compounds. Microalgae are also important for biotechnology and people have harnessed them to make food, fuel and medicines. Green microalgae live in many types of habitats from streams to oceans, and they can also be found on the land, including in deserts. Like plants that live in the desert, these microalgae have likely evolved specific traits that allow them to live in these hot and dry regions. Yet, fewer scientists have studied microalgae compared to land plants, and until now it was not well understood how microalgae could survive in the desert.

Nelson et al. analyzed green microalgae from different locations around the United Arab Emirates and found that one microalga, known as *Chloroidium*, is one of the most dominant algae in this area. This included samples from beaches, mangroves, desert oases, buildings and public fresh water sources. *Chloroidium* has a unique set of genes and proteins and grew particularly well in freshwater and saltwater. Rather than just harnessing sunlight, the microalgae were able to consume over 40 different varieties of carbon sources to produce energy. The microalgae also accumulated oily molecules with a similar composition to palm oil, which may help this species to survive in desert regions.

A next step will be to develop biotechnological assets based on the information obtained. In the future, microalgae could be used to make an oil that represents an alternative to palm oil; this would reduce the demand for palm tree plantations, which pose a major threat to the natural environment. Moreover, understanding how microalgae can colonize a desert region will help us to understand the effects of climate change in the region.

*2015*). Its oleogenic properties, robust growth, and capacity to survive in a wide range of environmental conditions prompted us to carry out whole-genome sequencing and phenomic analyses. Genomic (**Dataset 1**) and phenomic (**Dataset 2**) datasets are available online at Dryad (*Nelson et al., 2017*).

We discovered that this species accumulates palmitic acid that, in conjunction with other traits, may enable its survival in a desert climate. The mechanisms employed by desert extremophiles such as *Chloroidium sp. UTEX 3007* to maintain cellular integrity despite the oxidative insults of a desert climate may yield insight into the biology of a key player in a desert ecosystem and may also provide resources for the production of highly thermo-oxido-stable oils for human use.

One of the most important thermo-oxido-stable oils for human civilization is produced by the oil palm tree, *Elaeis guineensisis* (*Barcelos et al., 2015*). Palm oil's distinguishing characteristics stem from its uniquely high concentrations of palmitic acid, which, due to its carbon chain length and lack of reactive double bonds, remains stable in conditions that destroy other longer and more unsaturated fatty acids. However, cultivation of trees for palm oil production has led to extensive destruction of high-biodiversity rainforests and wetlands that threatens to expand around the globe (*Abrams et al., 2016*). In addition to the loss of unique and diverse flora and fauna (*Labrière et al., 2015*; *Wich et al., 2014*), clear-cutting these regions for palm oil plantations has devastated major carbon sinks (*Yue et al., 2015*) and caused intensive production of smoke pollution that affects millions of people in densely populated areas (*Bhardwaj et al., 2016*; *Vadrevu et al., 2014*). Thus, an alternative source of palmitic acid could be valuable in terms of elevated product output, environmental preservation and a reduction of smoke pollution.

## Results

We re-isolated *Chloroidium* strains from diverse locations around the UAE including coastal beaches, mangroves, and inland desert oases (*Figure 1a–b*). The frequent re-isolation of the species from these different habitats suggests a broad habitat range in the local, predominantly desert, region. Due to its ubiquitous occurrence and lipid accumulation phenotype (*Figure 1c*), we selected *Chloroidium sp. UTEX 3007* for study as a well-adapted species within the UAE and a potential candidate species of interest for production of oil replacement products (*Sharma et al., 2015*).

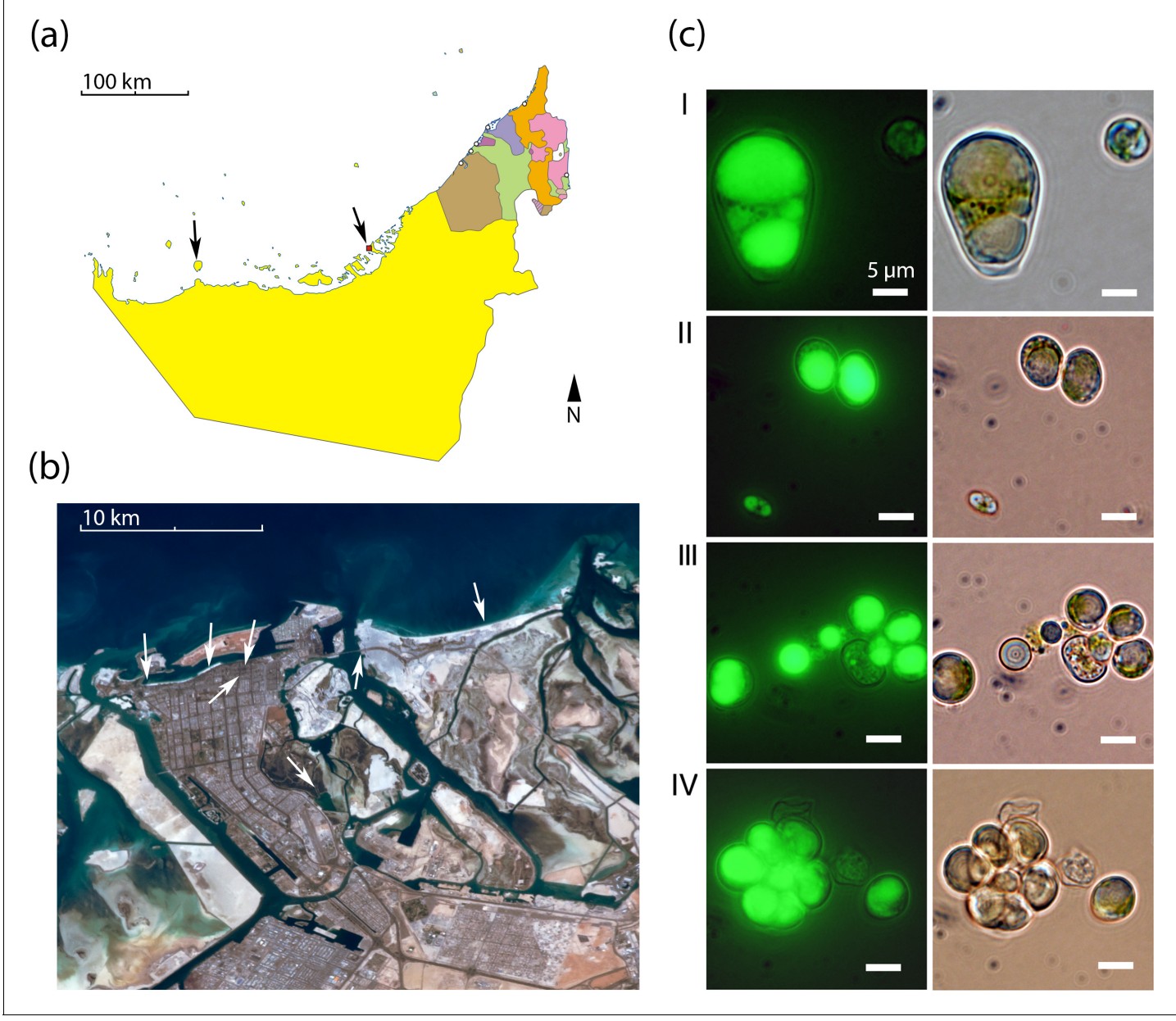

**Figure 1.** Geography and morphology of isolated *Chloroidium* strains. *Chloroidium* strains were isolated from samples taken from the indicated locations in (**a**) the UAE, and (**b**) within Abu Dhabi city specifically. *Chloroidium* strains were found in estuaries, mangrove forests, adhered to buildings, in municipal waters, etc. (**c**) *Chloroidium sp. UTEX 3007* stained for lipid bodies with BODIPY 505/515 and observed by fluorescence microscopy (left column) and phase-contrast microscopy (right column). (I) *Chloroidium sp. UTEX 3007* aplanospore displaying palmello-like morphology contrasted with a smaller vegetative cell, (II) two average-sized *Chloroidium sp. UTEX 3007* cells can be seen in the upper right of the panel while a miniature, recently hatched from an autospore, can be seen in the lower left, (III) Stained oil stores excreted from a *Chloroidium sp. UTEX 3007* cell, and (IV) a hatched, non-segregating autospore.

## Bioreactor growth

*Chloroidium sp. UTEX 3007* reproduces via unequal autospores (2–8 cells/division); cells vary greatly in size from 1 to 12 μm in diameter (mean = 6 μm; *Figure 2a*) and exhibit ovoid morphology, parietal chloroplasts, and accumulation of prominent intracellular lipid deposits (*Figures 1* and *2*). Stationary phase is reached after ~2 weeks of growth in F/2 media (*Figure 2*). *Chloroidium sp. UTEX 3007* showed robust growth in open pond simulators (OPSs), which generate a sinusoidal approximation

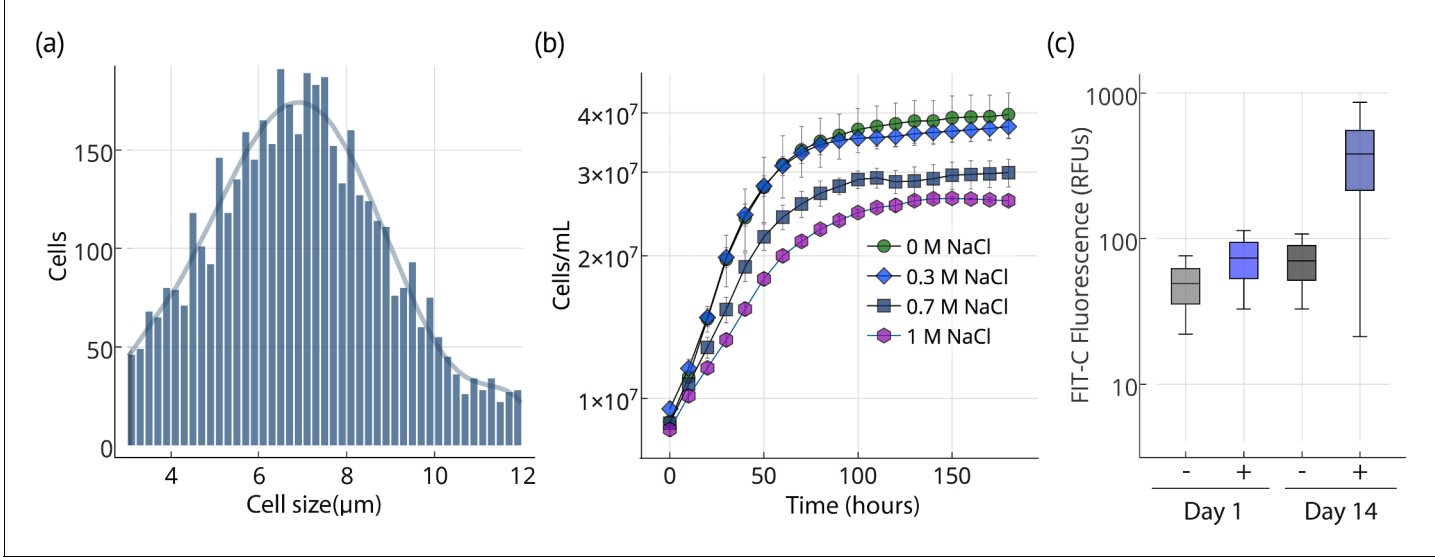

**Figure 2.** Cell size, growth, and lipid accumulation in *Chloroidium sp. UTEX 3007*. (a) Cell size distribution of *Chloroidium sp. UTEX 3007* in late log phase (14 days). Cell size analysis was performed using a Cellometer Auto M10 from Nexcelcom Bioscience (Lawrence, MA, USA) on 2 × 20 ul from each liquid algal sample. As autospores typically generate six progeny, the distribution was fitted with the sixth order polynomial equation: ($a + b*x + c*x^2 + d*x^3 + e*x^4 + f*x^5 + g*x^6$). (b) Growth of *Chloroidium sp. UTEX 3007* cultures in tris-minimal media supplemented with 0, 0.3, 0.7. and 1M NaCl at 20°C and 400 umol photons $m^{-m}$ $s^{-s}$ of full-spectrum light. Cells were grown in a Multi-Cultivator MC 1000 by Photon Systems Instruments (Drasov, Czech Republic. See *Figure 2—figure supplement 2* for chlorophyll/cell count curve and *Figure 2—figure supplement 1* for a growth curve in the open pond simulators (OPSs). (c) Fluorescence intensity of cells (RFUs=relative fluorescence units) measured with a BD FACSAria III flow cytometer (BD Biosciences, San Jose, CA) at mid-log phase (time=day 1, one week after inoculation) and one week into stationary phase (time=day 14) with and without staining (indicated as (+) or (-), respectively, from the lipophilic dye BODIPY 505/515. Whiskers indicate range, box edges indicate standard deviation and the box centerline indicates the median fluorescence intensity reading.

The following source data and figure supplements are available for figure 2:

**Source data 1.** Cell diameter measurements.
**Source data 2.** Cell concentration time course measurements.
**Source data 3.** Flow cytometry measurements.
**Figure supplement 1.** OPS growth curve.
**Figure supplement 2.** Chlorophyll/cell count curve.

of daily light based on geographical information system (GIS) data on daily light/dark cycles to simulate the growth conditions of an outdoor large-scale algae growth operation (*Figure 2—figure supplement 1*) (*Lucker et al., 2014*; *Tamburic et al., 2014*). We note that, because OPSs tend to underestimate growth rates compared to actual open ponds (*Lucker et al., 2014*; *Tamburic et al., 2014*), the actual productivity in an open pond may be higher. Under simulated parameters, growth occurs at 0–60 g/L NaCl and culture maturity (i.e., the transition from log to stationary phase) is reached after about two weeks (*Figure 2b*).

## Lipid accumulation

Upon entering stationary phase, lipid accumulation commences and causes an increase in size, weight (2-6x, ~100–200 pg/cell dry weight), and relative fluorescence of lipophilic dye stained cells (*Figures 1* and *2*). After reaching stationary phase, *Chloroidium sp. UTEX 3007* cultures remain stable and continue to increase biomass at a reduced rate (*Figure 2b*).

Our results indicate that the slowly accruing biomass in post-stationary phase is concurrent with an accumulation of triacylglycerols (TAGs) composed of mostly palmitic acid side-chains (*Figure 3a*). The presence of palmitic acid instead of other longer and more unsaturated fatty acids is expected to increase fitness at higher temperatures because palmitic acid is more thermostable.

## Intracellular metabolite profiling

We performed gas (GC-FID for fatty acids and GC-MS for polar primary metabolites) and liquid chromatography (UHPLC/Q-TOF-MS/MS) on cellular extracts of *Chloroidium sp. UTEX 3007* to characterize its intracellular metabolites. Fatty acids were extracted for gas chromatography with flame-ionization detection (GC-FID) analysis at one week after the onset of stationary phase. Fatty acid methyl esters were created to compare their abundance with a fatty acid methyl ester (FAME) standard library. Comparison with an internal standard (C15:0) revealed that fatty acids had accumulated to 78.2 ± 7.7% (from 8 replicates), and ~41.8% of total fatty acids consisted of palmitic acid at the time of harvest (*Figure 3a*). The palmitic acid content of *Chloroidium sp. UTEX 3007* was found to be higher than many other known algae (*Lang et al., 2011*) roughly equivalent to that of palm oil from *Elaeis guineensis* (*Barcelos et al., 2015*) (*Figure 3b*).

We performed UHPLC/MS-QToF to identify intact and distinct lipid species in *Chloroidium sp. UTEX 3007* and *Chlamydomonas reinhardtii* (CC-503) (*Figure 3c*). Microwave-assisted methanol whole-cell extracts were filtered and used for LCMS analyses (Dataset 2). *Chlamydomonas reinhardtii* contained more membrane-type lipids including sphingolipids and monogalactosyl diacylglycerides (MGDGs), which indicated that the accumulation of neutral lipids had not yet commenced under these conditions. *Chloroidium sp. UTEX 3007* contained a larger fraction of triacylglycerols at the conditions tested (3 weeks at 25°C, 400 μmol photons/m$^2$/s). In stationary phase *Cloroidium sp. UTEX 3007* cells, we observed a marked accumulation of mono-, di-, and triacylglycerides. These larger lipid molecules formed six notable peaks at the end of the chromatograms where the concentration of isopropanol in the elution solvent was between 70–90%. *Chloroidium sp. UTEX 3007* cultures grown for the same duration, under the same conditions, and having the same chlorophyll *a* content as *Chlamydomonas reinhardtii* contained a dramatically higher percentage of triacylglycerols in the final extracts (*Figure 3c*).

## Metabolic requirements

To define the spectrum of nutrients and metabolites that support heterotrophic growth, we exposed the strain to hundreds of chemical compounds in phenotype microarray plates (*Figure 4a*, **Dataset 2**). This method has been used successfully to identify new compounds that support heterotrophic growth in the model green alga *Chlamydomonas reinhardtii* (*Chaiboonchoe et al., 2014*). *Chloroidium sp. UTEX 3007* grew on more than 40 different carbon sources, including several desiccation-promoting sugars including trehalose, sorbitol, raffinose, and palatinose (**Dataset 2**).

Although *Chloroidium* species have been documented to accumulate the pentose ribitol, to date, no green alga has been shown to use a pentose sugar (for example arabitol and lyxose, which *Chloroidium sp. UTEX 3007* can assimilate) for heterotrophic growth. Thus, our experiments show that some green algae have a wider range of sugar utilization than previously recognized. Because *Chloroidium sp. UTEX 3007* was able to use common plant polysaccharides and sugars, including cellulose, fructose, and fucose for heterotrophic growth (*Figure 4*), our nutrient assays suggest a herbivoric capacity for the *Chloroidium* species. Strikingly, growth on L-glucose was also observed in our assays.

In contrast to *Chlamydomonas reinhardtii*, which can only use acetate as a carbon source for heterotrophic growth, *Chloroidium sp. UTEX 3007* lacked the capacity to assimilate extracellular acetate for growth. We found that a different profile of nitrogen sources supports growth in *Chloroidium sp. UTEX 3007* in comparison with *Chlamydomonas reinhardtii*. Although *Chlamydomonas reinhardtii* can use a broad range of nitrogen sources, including several D-amino acids (*Chaiboonchoe et al., 2014*), major nitrogen sources promoting growth of *Chloroidium sp. UTEX 3007* were limited to ammonium, nitrate, urea, agmatine, ornithine, and glutamine (*Figure 4a*). These differences in nutrient assimilatory capacities might reflect differences in the habitats of the two algae, such as the composition of the soil or other species in their immediate vicinity. A desert environment may offer only

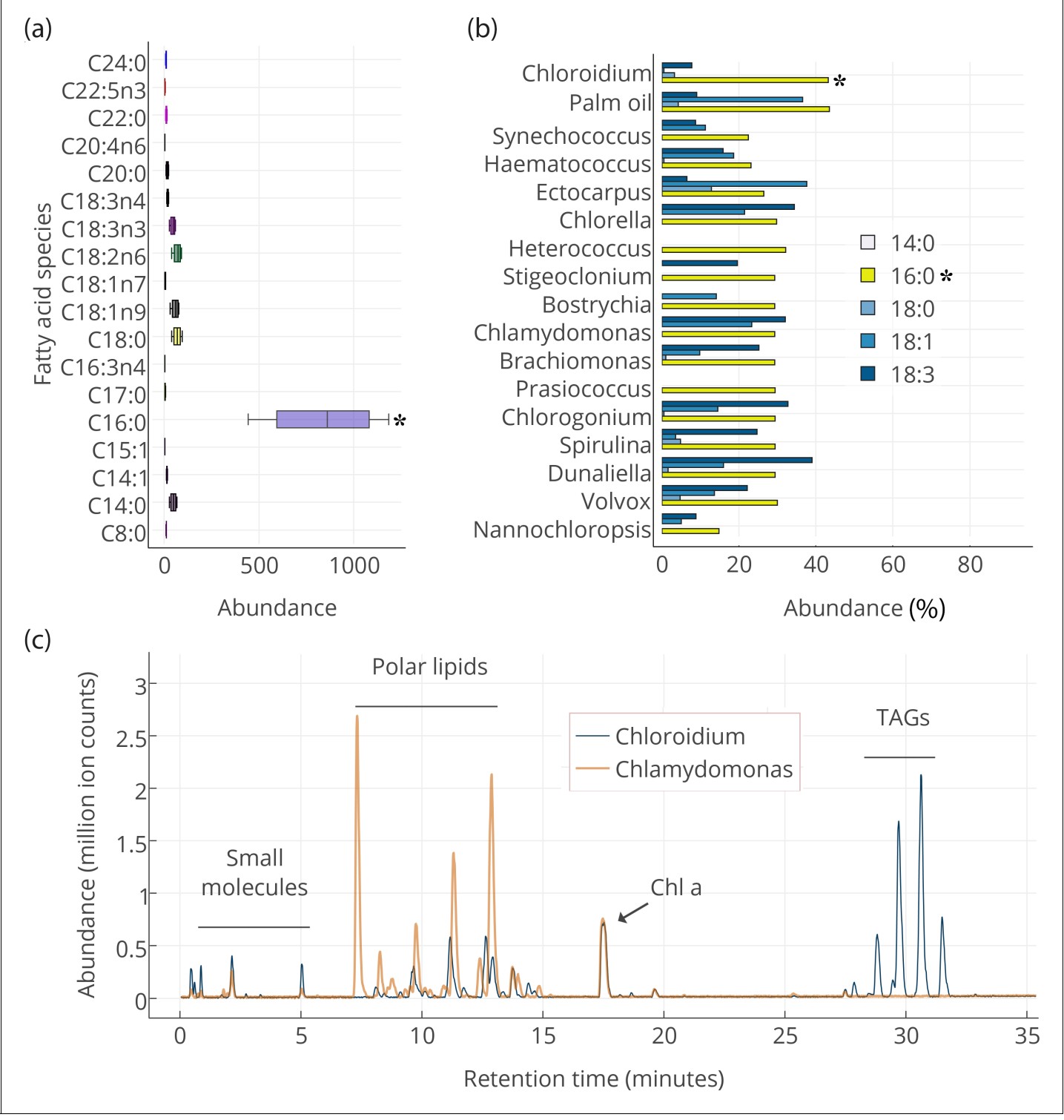

**Figure 3.** Lipid composition of *Chloroidium sp. UTEX 3007* observed with HPLC/MS and GC-FID, and comparison with other photosynthetic, oleagenic species. (a) *Chloroidium sp. UTEX 3007* fatty acid content estimated by extracting total lipid, creating methyl esters, and running the esters on a GC-FID (whiskers = range, box boundaries = 1 SD, center box line = mean [8 replicates]). (b) Comparison of fatty acid profile of *Chloroidium sp. UTEX 3007* with that of oil palm (*Elaeis guineensis*) (*Barcelos et al., 2015*) and several other algal isolates (*Lang et al., 2011*). Asterisks mark the presence of palmitic acid in *Chloroidium sp. UTEX 3007*. (c) Base peak chromatograms (BPCs) from stationary phase *Chloroidium sp. UTEX 3007* and *Chlamydomonas reinhardtii* extracts run in positive mode on an Agilent LC-MS QToF 6538 (Agilent, Santa Clara, CA, USA) using an acetonitrile/
*Figure 3 continued on next page*

*Figure 3 continued*

ammonium formate/isopropanol gradient. Cultures were grown for three weeks in F/2 media with 0 g/L NaCl (freshwater media). Triacylglycerols (TAGs) can be seen in abundance in *Chloroidium sp. UTEX 3007* while *Chlamydomonas reinhardtii* contained a higher ratio of polar lipids (Dataset 2).

The following source data is available for figure 3:

**Source data 1.** GC-FID results for major fatty acid species in *Chloroidium sp.* UTEX 3007.

**Source data 2.** Fatty acid profiles of *Chloroidium sp. UTEX 3007, Elaeis guineensis* (*Barcelos et al., 2015*), and several other algal isolates (*Lang et al., 2011*).

**Source data 3.** HPLC-MS base peak chromatograms (BPCs) for *Chloroidium sp. UTEX 3007* and *Chlamydomonas reinhardtii* extracts.

limited nitrogen sources, while the fertile soil of typical *Chlamydomonas reinhardtii* habitats, that is, the temperate climate zone, often contains a wide variety of nitrogen sources (*Harris, 2009*).

In addition to GC-FID and HPLC-MS, we also performed metabolite profiling using GC-MS to examine other simple carbon and nitrogen compounds in *Chloroidium sp. UTEX 3007* and *Chlamydomonas reinhardtii*. While *Chloroidium sp. UTEX 3007* accumulated a wide assortment of sugars, a much broader range of nitrogen metabolites was accumulated by *Chlamydomonas reinhardtii* (*Figure 4b*). These results complement the phenotype microarray results. Carbon compounds accumulated by *Chloroidium sp. UTEX 3007* include the desiccation resistance-promoting sugars arabitol, ribitol, and trehalose (*Figure 4b*). Based on previous findings, (*Santacruz-Calvo et al., 2013*; *Watanabe et al., 2016*), these sugars and lipids are likely to be involved in osmotic stabilization in *C. sp. UTEX 3007*.

## Genomic analysis

We sequenced and annotated the genome of *Chloroidium sp. UTEX 3007* at high depth (~200 x) with PCR-free Illumina reads to yield a 52.5 Mbp genome (N50 = 148 kbps). Although we did not perform classical chromosome quantification, we estimate that *Chloroidium sp. UTEX 3007* has approximately 16 nuclear chromosomes based on synteny analyses performed with *Coccomyxa subellipsoidea* C-169 (20 chromosomes) and *Chlorella variabilis* NC64A (12 chromosomes). *Chlorella*-type ($G_3T_3A$) repeat telomeres were discovered on nuclear contigs. Protein-coding sequences, comprising 40.0% of the genome, contained 9455 distinct Pfam domains (Dataset 1, within 8155 genes) as predicted with Pfam-A.hmm (v31.0). Gene predictions using an *Arabidopsis thaliana* hidden Markov model (HMM) in SNAP (*Korf, 2004*) were used for downstream analyses (Dataset 1).

Sequence similarity analysis of the predicted proteome to known species revealed matches at or below E-values <$10^{-10}$ with proteins from species of *Coccomyxa* and *Chlorella* (Dataset 1). We hypothesize that *Chloroidium sp. UTEX 3007* is epiphytic in nature, and that such an ecological niche is responsible for the broad carbon assimilatory behavior we observed in both the (1) phenotype microarrays and (2) reconstructed genetic pathways from genomic data. The retention of these sugar-inclusive metabolic pathways may provide survival advantages in a desert climate due to the osmotic stability and desiccation tolerance conferred by some of the produced sugars.

## Genome-scale metabolic network reconstruction and oil accumulation pathways

From our functional annotation of *Chloroidium sp. UTEX 3007*'s genome, we reconstructed a genome-scale metabolic pathway model by mapping predicted genes to metabolic network nodes reconstructed from pathways in the BioCyc database (https://biocyc.org/). The model can be found in Dataset 1. Our metabolic model highlights known pathways of potential lipid biosynthesis in *Chloroidium sp. UTEX 3007*. This analysis also revealed unusual lipid biosynthesis mechanisms including TAG biosynthesis from membrane lipids instead of an acyl-CoA pool. This has previously been documented in some yeast and plant species that use phospholipids as acyl donors and diacylglycerol as the acceptor (*Cases et al., 2001*; *Dahlqvist et al., 2000*).

To investigate possible pathways in *Chloroidium sp. UTEX 3007* that could synthesize TAGs from polar membrane lipids, we examined enzymes with phospholipid-cleaving or phospholipid-

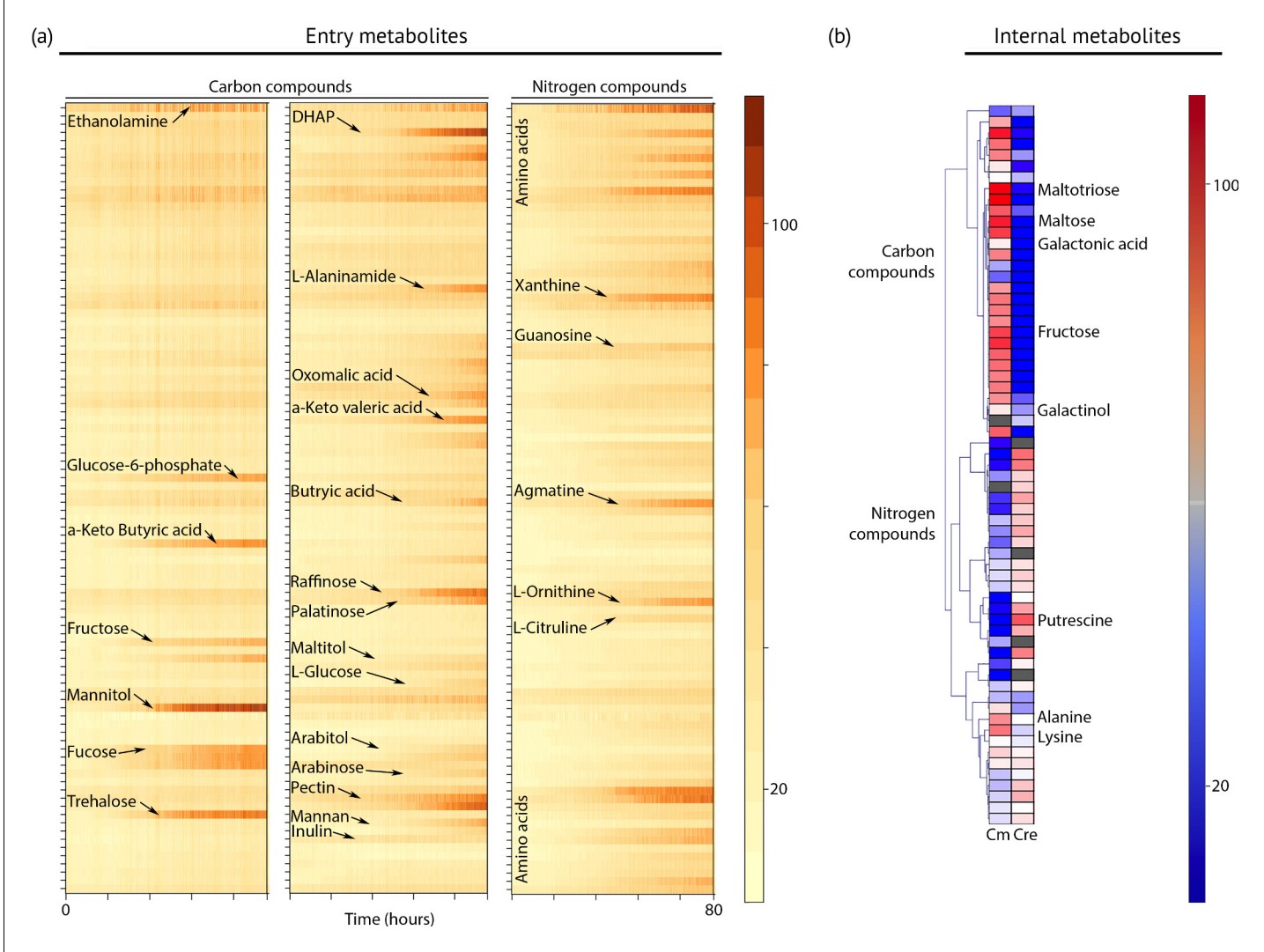

**Figure 4.** Metabolic profiling of *Chloroidium sp. UTEX 3007*. (**a**) Biolog phenotype microarrays were run using an Omnilog instrument (Biolog Inc., Hayward, USA) as previously described (*Chaiboonchoe et al., 2014*). In total, 380 substrate utilization assays for carbon sources (PM01 and PM02 plates), 95 substrate utilization assays for nitrogen sources (PM03 plate), 59 nutrient utilization assays for phosphorus sources, and 35 nutrient utilization assays for sulfur sources (PM04 plate), along with peptide nitrogen sources (PM06-08 plates) were performed (Dataset 2). All microplates were incubated at 25°C for up to 8 days, and the dye color change (in the form of absorbance) was read with the Omnilog system every 15 min. As the Omnilog instrument does not provide a source of continuous light during incubation, the algae are assumed to be carrying out heterotrophic respiration. In addition, a marked increase in the chlorophyll *a* content and total cell count was confirmed for wells with suggested growth. Kinetic curves were plotted from the raw data in the form of heatmaps, and statistical analysis was carried out to visualize the metabolic properties and generate Omnilog values. Heatmap density correlates to Omnilog-registered color density. In addition to the dye color change, Omnilog also registers color change resulting from the accumulation of other pigments, including chlorophyll *a*. Thus, growth is displayed as cumulative color change density. (**b**) Extraction and analysis by gas chromatography coupled with mass spectrometry was performed as described in *Lisec et al. (2006)*. The color scale corresponds to chromatogram peak areas reported in the GC-MS results in Dataset 2. Significant increase of diverse carbon compounds was observed in *Chloroidium sp. UTEX 3007* (Cm) as compared to *Chlamydomonas reinhardtii* (Cre), and vice versa for nitrogen compounds. These differences may reflect acclimatization to their respective habitats and lifestyles.

The following source data is available for figure 4:

**Source data 1.** Phenotype microarray results for plates PM1, PM2, and PM3.

**Source data 2.** GC-MS results for *Chloroidium sp. UTEX 3007* (Cm) and *Chlamydomonas reinhardtii* (Cre) intracellular polar metabolites.

translocating activities. Our BLASTP/BLAST2GO analyses highlighted an abundance of phospho-lipid-translocating ATPases that might be involved in this process (Dataset 1). From our protein family domain analysis, we detected lecithin retinol acyltransferase (LRAT) and phospholipase D (PLD) domain-containing enzymes in *Chloroidium sp. UTEX 3007* (*Figure 5*) that are also candidate proteins for rapidly modifying membrane lipids (Dataset 1). Membrane-bound LRAT enzymes are involved in the transfer of palmitoyl groups to and from a variety of biomolecules (*Furuyoshi et al., 1993*; *Golczak and Palczewski, 2010*), and several phospholipases, including those from the PLD family, cleave phosphate groups from membrane lipids. One predicted PLD in *Chloroidium sp. UTEX 3007* contains several C2 calcium-binding domains, indicating that it acts on membrane phospholipids to produce membrane-bound phosphatidic acid (PA) and cytosolic choline (*Rahier et al., 2016*). PA has noted roles in environmental stress response (*Peppino Margutti et al., 2017*), and its liberation has been shown to create 'supersized' lipid droplets (*Fei et al., 2011*). We hypothesize that phospholipases acting on membrane lipids could play particularly important roles in osmotic stress resistance and lipid accumulation in *Chloroidium sp. UTEX 3007* as they represent a key branch point between polar and non-polar lipid species.

## Unique protein families

We sought to compare and contrast *Chloroidium sp. UTEX 3007*'s genome with other species from diverse clades within the green algae. Our approach compares the euryhaline *Chloroidium sp. UTEX 3007* with salt- and fresh-water-dwelling species and highlights Pfam domains that are unique to *Chloroidium sp. UTEX 3007*. The genomes of *Chlorella variabilis NC64A* (*Blanc et al., 2010*), *Micromonas pusilla* (*Worden et al., 2009*), *Ostreococcus tauri* (*Blanc-Mathieu et al., 2014*), and *Coccomyxa subellipsoidea* (*Blanc et al., 2012*) were downloaded from the National Center for Biotechnology Information (NCBI), and the *Chlamydomonas reinhardtii* genome (*Merchant et al., 2007*) was downloaded from Phytozome (v5.5; www.phytozome.net). De novo gene prediction using an *A. thaliana* HMM in SNAP yielded a set of peptide predictions (Dataset 1) that served as a base for HMM alignment with Pfam-A (v31.0, (http://hmmer.org)).

*Figure 6* provides a graphical representation of unique and shared Pfam domains among the six algal species in an interactive Venn diagram (*Figure 5—source data 1*) using InteractiVenn (http://www.interactivenn.net/)(*Heberle et al., 2015*). *Chlorella sp. NC64A* contained the lowest number of unique Pfam domains (174), while, surprisingly, *Ostreococcus tauri*, known for its minimal genome, had the highest number of unique Pfams (427).

In comparison with other algae, we found 235 Pfam domains that were unique to *Chloroidium sp. UTEX 3007* and may play roles in its inhabitation of a desert region. Selected, unique, Pfam entries with relevance to phenotypes documented in this manuscript and unique to *Chloroidium sp. UTEX 3007* are shown in *Table 1*. For example, the detection of a NST1 domain suggests the presence of

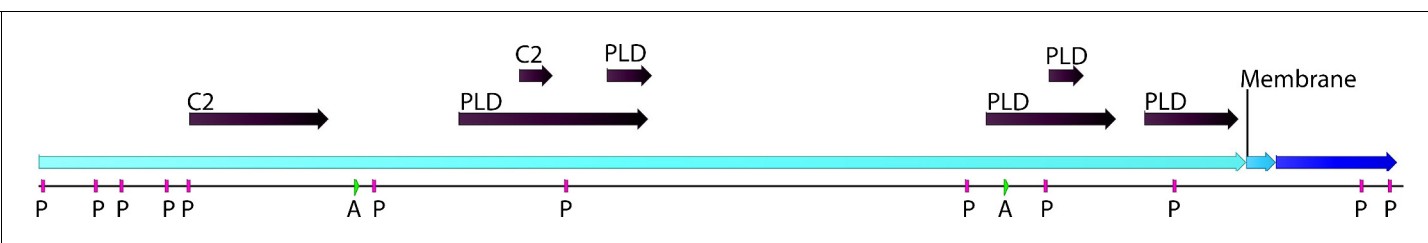

**Figure 5.** Phospholipase D (PLD; EC number: 3.1.1.4) domain-containing gene in *Chloroidium sp. UTEX 3007*. Functional protein family domains in a *Chloroidium sp. UTEX 3007* predicted protein with high similarity to PLD proteins from closely related organisms. PLDs are members of the phospholipase superfamily and produce phosphatidic acid as a main product. Phosphatidic acid is involved in signaling, membrane curvature, and is rapidly converted to diacylglycerol. The *Chloroidium sp. UTEX 3007* PLD contains several domains with calcium-binding, amidation, and phosphorylation sites (annotated as C2 (2), A (2), or P(11) domains). C2 domains act to target proteins to cell membranes and allow phosphatases to de-phosphorylate membrane lipids without removing them from the membrane.

The following source data is available for figure 5:

**Source data 1.** Locations, confidence scores, and accession numbers for PLD and C2 Pfam domains in *Chloroidium sp. UTEX 3007*.

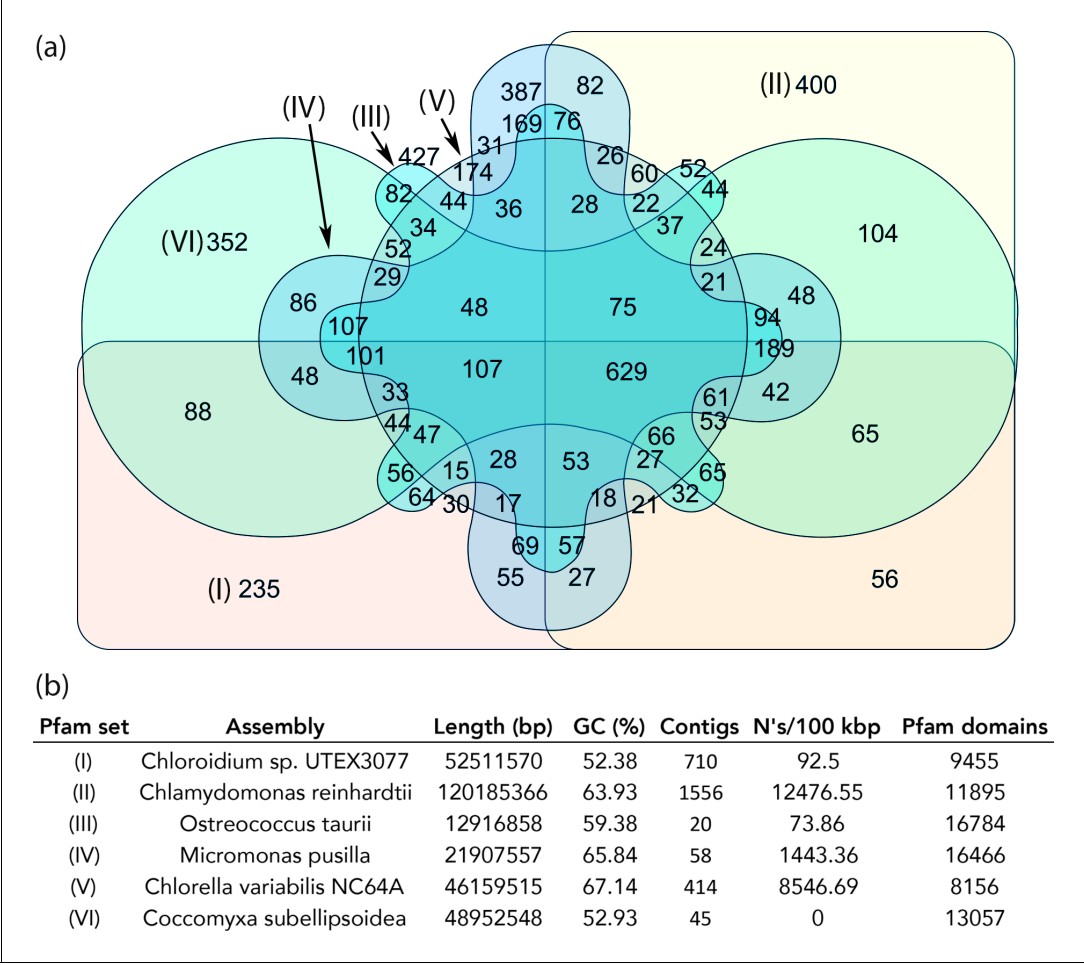

**Figure 6.** Protein family (Pfam) domains in *Chloroidium sp. UTEX 3007* compared with algae from other clades. Well-curated assemblies from genomes of algae from four other clades were downloaded from NCBI [assemblies - *Micromonas pusilla*, *Ostreococcus taurii*, and *Coccomyxa subellipsoidea*]. The *Chlamydomonas reinhardtii* genome was downloaded from Phytozome (v5.5) (**Merchant et al., 2007**). De novo gene prediction, including exon-intron structural modeling, yielded a set of peptide predictions that served as a base for HMM alignment with Pfam-A (v31.0, (http://hmmer.org).). (**a**) Shared and unique Pfam sets from representative species of major green algae clades. Pfam lists can be viewed by selecting the numbers displayed for each shared or unique value in *Figure 6—source data 3*. (**b**) Legend for (**a**) including additional metrics describing the algal assemblies used for de-novo coding sequence and Pfam predictions.

The following source data is available for figure 6:

**Source data 1.** Predicted Pfam designations for each species in *Figure 6*.
**Source data 2.** Table with QUAST results used in (b).
**Source data 3.** Interactive Venn diagram that can be viewed at interactivenn.net to obtain Pfam sets for numbers displayed in *Figure 6*.

proteins involved in cellular response to hypo-osmolarity (**Leng and Song, 2016**). Similarly, the presence of a proline-rich domain is associated with thermotolerance (**Cvikrová et al., 2012**; **Khan et al., 2013**). In all other cases for the abiotic stress column, oxidative stress resistance, either direct or metal-mediated, is suggested. Because mechanisms to deal with oxidative stress from a variety of sources (sun, salt, drought, heat) are necessary to survive in a desert climate, gene products encoded with these domains may play a key role in the successful colonization of a desert region, and its various sub-habitats, by *Chloroidium sp. UTEX 3007*.

**Table 1.** Protein families (Pfams) with roles in (a) abiotic stress resistance and (b) saccharide metabolism unique to Chloroidium among the green algae explored. Pfam domains were predicted using HMMsearch against the Pfam-A (v31.0, http://hmmer.org) database. The i-Evalue or the 'independent E-value', signifies the E-value that the query would have received if it were the only domain, irrespective of homology with any other entries in the database thereby providing a stringent confidence metrics for the hit (http://hmmer.org).

| | Description | Key | i-Evalue |
|---|---|---|---|
| (a) | Iron-containing redox enzyme | Haem_oxygenas_2 | 7.50E-05 |
| | Salt tolerance down-regulator | NST1 | 0.00012 |
| | Proline-rich | Pro-rich | 0.00015 |
| | Peroxisome biogenesis factor | PEX-2N | 0.003 |
| | OsmC-like protein | OsmC | 0.0066 |
| | NA,K-Atpase interacting protein | NKAIN | 0.003 |
| | Metallothionein | Metallothio_2 | 0.0074 |
| | Mercuric transport protein | MerT | 0.0054 |
| (b) | Beta-galactosidase jelly roll domain | BetaGal_dom4_5 | 0.013 |
| | Cryptococcal mannosyltransferase | CAP59_mtransfer | 3.50E-15 |
| | Carbohydrate binding domain CBM49 | CBM49 | 0.0085 |
| | Carbohydrate binding domain (family 25) | CBM_25 | 0.0023 |
| | Carbohydrate binding domain | CBM_4_9 | 0.00036 |
| | Cellulose biosynthesis protein BcsN | CBP_BcsN | 0.0018 |
| | Glycosyl hydrolase family 70 | Glyco_hydro_70 | 0.00054 |
| | Glycosyl hydrolase family 9 (heptosyltransferase) | Glyco_transf_9 | 0.0025 |
| | Carbohydrate esterase/acetylesterase | SASA | 2.00E-06 |
| | Activator of aromatic catabolism | XylR_N | 0.011 |

## Manganese catalase-like genes

BLAST2GO analysis revealed that *Chloroidium sp. UTEX 3007* contains a disproportionately high number of genes involved in redox chemistry and metal binding (Dataset 1) Exploring these genes further with functional and phylogenetic analyses highlighted a cluster of genes (CDS1/2/3) with homologs in desiccation-associated proteins from *Deinococcus* species (*Figure 7*, *Figure 7—figure supplement 1*).

The proteins from (CDS1/2/3) contain ferritin-like domains and may be involved in neutralizing reactive oxygen species (ROS) via manganese redox cycles (*Daly, 2009*; *Fredrickson et al., 2008*). Consistent with the primary sequence analysis, homology-based tertiary structure modeling confirmed that these proteins contain the metal binding motifs and extended C-terminal tails characteristic of manganese (Mn) catalases (*Whittaker, 2012*) (*Figure 7*). As intracellular iron can become toxic in the face of overwhelming oxidative stress from changing environments, the utilization of manganese as a replacement would offer a protective advantage in the face of severe oxidative stress (*Daly, 2009*; *Fredrickson et al., 2008*).

## Discussion

Algal genomes are under-represented among sequenced organisms (*Grossman et al., 2010*; *Umen and Olson, 2012*), and genome-scale studies fully incorporating metabolomics are, in general, poorly represented within the scientific literature (*Chaiboonchoe et al., 2014*). As microphytic species play fundamental roles in maintaining nutrient flow through ecosystems (*Akhter et al., 2016*), the study of their genomic and phenotypic characteristics is critical to fully understand local and global ecosystem dynamics. Our analysis provides a detailed genome-phenome description of a

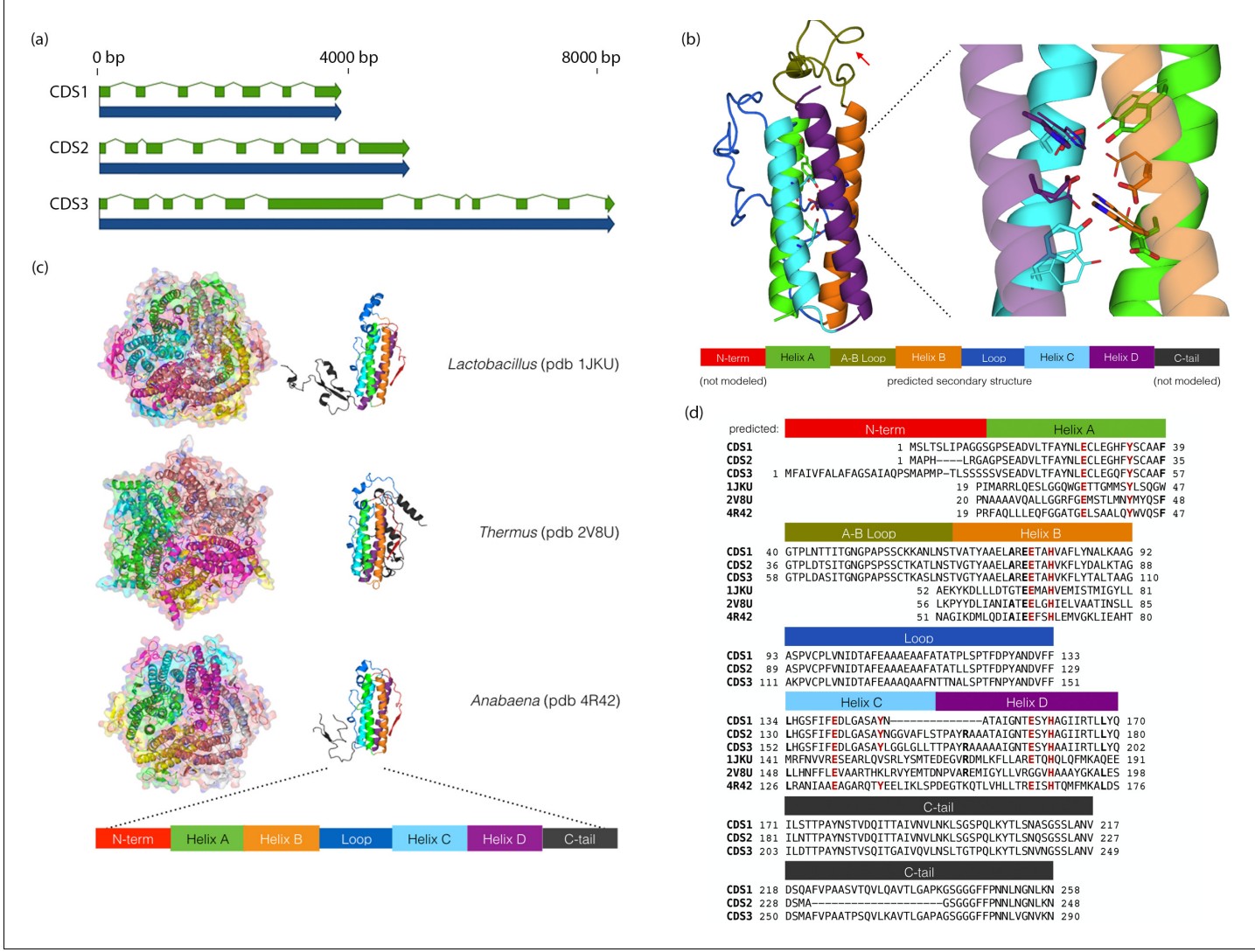

**Figure 7.** *Chloroidium sp. UTEX 3007* genes exhibiting homology with desiccation-responsive *Deinococcus* genes. (**a**) Gene models for the putative manganese catalases (CDS1/2/3). Functional annotation of homologs from (CDS1/2/3) is limited: 96 of the top 100 BLAST hits of CDS3 in the non-redundant database (nr v. 2.31, NCBI) are annotated as hypothetical or predicted proteins. Of the other four, three are characterized as desiccation related (from *Auxenochlorella protothecoides*, *Deinococcus gobiensis I-0*, and *Salinisphaera shabanensis E1L3A*). (**b**) Experimental structures of representative Mn catalases (*Antonyuk et al., 2000*; *Barynin et al., 2001*; *Bihani et al., 2013*) showing the homohexamers (left) and isolated monomers (right). The latter are colored according to structural motifs as given. (**c**) Left: Homology model of CDS3, colored according to linear structure shown below (the additional presumptive loop is indicated by the red arrow and the predicted Mn coordinating side chains are shown as sticks). Right: Zoom in of same homology model aligned with the metal coordinating residues of *Anabaena PCC 7120* Mn catalase (pdb 4R42); highlighting the spatial alignment of side chains (homology model side chains shown as thick sticks, 4R42 as thin sticks). (**d**) Sequence alignment of CDS1-3 shown with the predicted secondary structure. The presumptive helical regions are also aligned with the corresponding sequences from the known Mn catalase structures. Canonical Mn coordinating residues are in red and other positions with sequence identity between CDS1-3 and at two least of the known catalases are in bold.

The following source data and figure supplement are available for figure 7:

**Source data 1.** Models of CDS1-3 in protein data bank (PDB) format.

**Source data 2.** Amino acid residue alignment of CDS1-3.

**Figure supplement 1.** Alignment of CDS3 to a *Deinococcus globiensis* desiccation-related protein (*Yuan et al., 2012*).

green alga from a desert area and reveals insight into traits that could facilitate acclimatization to this climate.

## Fitness-enhancing traits and practical implications

Halo-tolerant algae are in demand for the cost-effective commercial production of a variety of compounds (*Chokshi et al., 2015*). Euryhaline algae such as *Chloroidium sp. UTEX 3007* allow algae cultivators to use local water sources including lakes, inland seas, or oceans. As fresh water is a scarce resource, the ability to produce commercial products photosynthetically by mass culture of algae using alternative water sources, including salt water, could lead to a reduced environmental footprint in the mass culture of algae.

We found that *Chloroidium sp. UTEX 3007* accumulated high levels of palmitic acid and various sugars (*Figures 1–4*). As growth and cell cycle progression involves the global synthesis and breakdown of an array of proteins and signaling molecules, the energy that the cell would usually spend in cell cycle progression and for biomass production during log phase might funnel directly into the accumulation of various carbon compounds in stationary phase. Chlorophytes are known to assume cystic morphology to survive desert conditions, and completely desiccated algae have been shown to survive until they reach nutrient-rich environments (*Abe et al., 2014*). The retention of chlorophyll a well into stationary phase indicates that *Chloroidium sp. UTEX 3007* cells may transition into a carbon storage state wherein cells photosynthetically fix carbon as resting autospores.

We propose that *Chloroidium sp. UTEX 3007* accumulates stable hydrocarbons such as palmitic acid to survive the extended periods of heat, drought and nutrient depletion that are characteristic of desert regions. As an example of the potential importance of palmitic acid in thermotolerance, a mutant of Arabidopsis deficient in palmitic acid desaturation had an increased optimal growth temperature range (*Kunst et al., 1989*).

Likewise, the various sugars that accumulated in *Chloroidium sp. UTEX 3007* have been shown to promote desiccation tolerance in a wide variety of species (*Bradbury, 2001*; *Breeuwer et al., 2003*; *Petersen et al., 2008*; *Petitjean et al., 2015*; *Santacruz-Calvo et al., 2013*; *Watanabe et al., 2016*). Uncommon sugars are known to be involved in the stress responses of green algae, but their constituents vary widely between even closely related species (*Gustavs et al., 2010*). For example, a study focusing on sugars in aeroterrestrial algae found sorbitol in the Prasiola clade, ribitol in the Elliptochloris and Watanabea clades, and erythritol in *Apatococcus lobatus* sap (*Darienko et al., 2015*; *Gustavs et al., 2010*), the expanded carbon source utilization capacity we observed in our phenotype microarray experiments supports the sequence data in placing this isolate within the herbivoric/saprophytic/aeroterrestrial Watanabea clade of green algae (*Darienko et al., 2010*). The proximity of *Chloroidium sp. UTEX 3007* isolation sites with date palm trees (*Phoenix dactylifera*) and mangrove species (*Avicennia marina*) indicates that these desert- and saline-hardy trees may be hosts for *Chloroidium sp. UTEX 3007*.

An anomaly in the natural world, heterotrophic growth on L-glucose could further indicate a close relationship with plants. Of the very few species documented to use L-glucose for growth, two species, one bacterium and one fungal species, live endo-/epi-phytic lifestyles (*Sasajima and Sinskey, 1979*; *Shimizu et al., 2012*). L-glucose biosynthesis in a plant system was characterized in the 1970's (*Barber, 1971*). Barber concluded that L-glucose biosynthesis proceeds via an epimeration of the D-mannose moiety of guanosine 5'-diphosphate D-mannose to the L-galactose of $\beta$-L-galactose 1-phosphate. Our data suggest that the uptake of L-glucose may proceed through a mannose-nucleotide intermediate (*Lunn et al., 2014*).

## Genomics of Chloroidium

The genome of *Chloroidium sp. UTEX 3007* revealed many potential cellular mechanisms for dealing with osmotic stress (*Figure 3*, *Table 1*). Due to the high temperatures encountered in its natural habitat, the presence of longer unsaturated fatty acids may be detrimental to membrane stability and greater quantities of stable fatty acids such as palmitic acid may be required. As unsaturated lipids are at greater risk for peroxidation upon salt stress (*Shu et al., 2015*), an increase in saturated membrane lipids can also be protective against rapid shifts in salinity. Our observations suggest that proteins involved in salt stress response can translate both into heat stress tolerance and increased lipid accumulation due to their release of PA from membrane lipids. As increased PA has been shown to

cause 'supersized' lipid droplets (*Fei et al., 2011*), and salt stress promotes lipid storage (*Wang et al., 2016*), the accumulation of lipids in response to salt stress may be occurring through the action of PLD. Several studies have found that the overexpression of transgenic PLD confers salt and drought tolerance in plants, however a trans gene from a euryhaline organism has not yet been tested (*Ji et al., 2017*; *Wang et al., 2014*; *Yu et al., 2015*). As such, the PLD from the euryhaline *Chloroidium* might present an attractive target for biotechnology efforts.

The stores of palmitate we observed may play other roles than energy storage or resistance to heat and salt stress. A pool of palmitic acid may be necessary for dynamic palmitoylation of membrane-docked ion pumps. Palmitoylation, mediated by LRAT, can increase targeted regional hydrophobicity and is an expected requisite for a massive influx of ion channels into a cell membrane. However, although at least three subunits of the human cardiac sodium pump are regulated by palmitoylation, the full functional outcome of these palmitoylation events is still not well characterized (*Howie et al., 2013*). The increased targeting of ion pumps into *Chloroidium sp. UTEX 3007* membranes by palmitoylation could also assist in surviving rapid salinity shifts and hypersaline conditions.

*Chloroidium sp. UTEX 3007* appeared to have an especially sophisticated network of genes involved in metal metabolism. As metal-containing enzymes have important roles in oxidative stress reduction and in protein folding in the face of severe environmental conditions, these genes may promote survival in a desert region. Daly and colleagues have proposed that Mn redox cycles are a critical mechanism by which desiccation and radiation tolerant organisms avoid damage caused by reactive oxygen species (ROS), which can be generated by spurious Fe-catalysis (*Daly, 2009*; *Fredrickson et al., 2008*). An intermediate protective step to prevent ROS damage is the decomposition of hydrogen peroxide, as accumulated hydrogen peroxide can react with free iron via Fenton chemistry to generate extremely reactive hydroxyl radicals (*Daly, 2009*; *Fredrickson et al., 2008*). The radiation tolerance of organisms shows a positive correlation with a higher intracellular Mn to Fe ratio. Moreover, Mn catalase overexpression provides oxidative protection in cyanobacteria (*Fenton, 1894*), and homologs of these genes are upregulated in response to desiccation stress in other *Chlorella* strains (*Gao et al., 2014*). Thus, the cumulative evidence suggests that Mn catalases in *Chloroidium sp. UTEX 3007* and other species could be used to mitigate ROS and cellular damage from environmental stressors.

## Outlook on remediation for palm oil production

The focus of our study, *Chloroidium sp. UTEX 3007*, grows robustly in a broad spectrum of conditions and accumulates palmitic acid. Thus, it may serve as an alternative source for this valuable fatty acid. The demand for palmitic acid has caused public pressure for the increased cultivation of oil palm trees (*Abrams et al., 2016*). However, the traditional cultivation and harvest of palm oil involves environmentally damaging practices that destroy high-biodiversity rainforests and produce large volumes of smoke pollution (*Newbold et al., 2015*). Although Southeast Asia is now the primary source of palm oil, new plantations are rapidly growing in Africa and Central America at the expense of rainforests that are no less diverse or unique than those of Asia (*Barcelos et al., 2015*). Recent studies have shown that transformation of native rainforest land causes approximately 50% reduction in species diversity, density, and biomass in animal communities (*Yue et al., 2015*). This is a critical global issue, as the pace of deforestation has accelerated dramatically in recent years (*Newbold et al., 2015*). The development of an alternative method for producing palm oil substitutes is therefore highly valuable, but thus far no viable option has been found. We envision that our characterization of *Chloroidium sp. UTEX 3007* and its associated genome and phenome information can be used to develop resources for the sustainable production of palm oil alternatives.

# Materials and methods

## Available datasets

Figure supplements and source data can be found online at Dryad (*Nelson et al., 2017*).

## Genome assembly and annotation

*Chloroidium sp. UTEX 3007* was sequenced with the Illumina HiSeq 2500 (Illumina, San Diego, USA) as described previously (*Sharma et al., 2015*). We performed an assembly with the CLC Genomics

Workbench assembler (v8.5, CLC Genomics, Qiagen, Aarhus, Denmark) with a kmer length of 45, word size of 22. After assembly, reads were mapped back to contigs and contigs were updated according to the mapped reads (95% sequence identity required). The insert size for paired genomic reads (2 × 100 bp) was ~720 bp on average and reached a maximum of 1200 bp. The reads assembled into 710 scaffolds with N50 of 150.8 kbp and a total length of 52.5 Mbp. For mapping, 335,693,730 reads were obtained of which 323,780,836 (98.6%) reads were matched to the final assembly (Dataset 1).

Using an *Arabidopsis thaliana* HMM in SNAP (*Korf, 2004*) to predict gene models, we predicted 42,000 individual transcripts and their translated protein amino acid sequences. Kingdom-specific HMM libraries have been found to increase the sensitivity and accuracy of genome annotations (*Alam et al., 2007*), and we found that using the *A. thaliana* HMM as a chlorophyte-specific HMM yielded the best gene models (in terms of intact Pfam domains) for *Chloroidium sp. UTEX 3007* as well as algae from several other lineages. The assemblies of ther algae used in the manuscript for comparative analyses (*Figure 6* and *Table 1*) were analyzed with QUAST and the results are presented in *Figure 6*. The full sets of resulting annotations are available as transcripts and proteins in fasta (.fa) format and GFF (.gff) annotation files (Dataset 1). All assemblies used in this manuscript were analyzed by us using QUAST (*Gurevich et al., 2013*).

We used the predicted genes/transcripts/proteins from SNAP (default parameters) for functional annotation, metabolic network reconstruction, and comparative analyses. The predicted proteins were analyzed for Pfam domains (*Rodrigues et al., 2015*) using HHMER (*Sinha and Lynn, 2014*) for protein family analysis (E-value <0.01, Pfam-A.hmm (v31.0, current) [*Finn et al., 2014*]) and analyzed for similarity to known proteins using BLASTP (v2.2.31) to create reference tables for annotated gene functions (Dataset 1).

## Metabolic network reconstruction

A model of the *Chloroidium sp. UTEX 3007* metabolic network was reconstructed from functional gene annotation and EC (enzyme code) evidence from BLASTP (Dataset 1) searches using the BLAST2GO v1.1.0 functional annotation tool (https://www.blast2go.com/) (*Jones et al., 2014*), which includes profile-based searches, to obtain model component genes (Dataset 1). The curated list of genes was processed using Pathway Tools v19.0 (*Karp et al., 20092010*) software. The pathologic function of Pathway Tools generated the genome-scale metabolic pathway model by matching our EC assignments with known enzyme/reaction references of MetaCYC (*Caspi et al., 2014*). Enzymes that were deleted or modified by the enzyme commission were manually updated. A total of 1400 proteins associated with 1448 genes, of which, 441 were associated with more than one EC number. The final model has 229 pathways, 1651 enzymatic reactions, 17 transport reactions, 1464 polypeptides, 22 transporters, and 1245 compounds.

PathoLogic assigned pathways include the following classes: activation/inactivation/interconversion (3), biosynthesis (165), degradation/utilization/assimilation (72), detoxification (3), generation of precursor metabolites and energy (21), metabolic clusters (5), and superpathways (30). We assigned 19 transport reactions, in which 15 are transport-energized by ATP hydrolysis. The number of genes, enzymes, and metabolites are comparable to a well-curated and verified reconstruction of the *Chlamydomonas reinhardtii* metabolic network, iRC1080 (*Chang et al., 2011*) and its recent expansion, iBD1106 (*Chaiboonchoe et al., 2014*), which includes 1106 genes and 1959 individual metabolites. Pathway classes and corresponding reactions that were shared between the *Chlamydomona s reinhardtii* iRC1086 Cobra model and the *Chloroidium sp. UTEX 3007* model (CF-SNAP) were: activation/inactivation/interconversion (3/0/2), superpathway (7), generation of precursor metabolites and energy (18), degradation/utilization/assimilation (28), biosynthesis (963), palmitate biosynthesis (53) and detoxification (5). The pathways/reactions unique to the *Chloroidium sp. UTEX 3007* model consists of: biosynthesis (796), generation of precursor metabolites and energy (18), degradation/utilization/assimilation (28), biosynthesis (462), palmitate biosynthesis (31), detoxification (6).

## Protein structure analysis

The Pfam database categorizes the CDS1-3 protein sequences (*Figure 7*) in the Ferritin_2 family (*Finn et al., 2014*). The structural core of ferritins and ferritin-like proteins is 4-helix bundle (with the pattern: helix A – hairpin- helix B – long crossover loop – helix C- hairpin – helix D) (*Andrews, 2010*).

The center of the helical bundle often contains a di-metal binding site and is frequently an active site for various forms of redox chemistry. For example, in canonical ferritins, this is a di-iron site that oxidizes $Fe2+$ to $Fe3+$ for mineral iron storage. Examination of the sequence indicated the proteins contain metal-binding motifs (helix A[E(6)Y], helix B[ExxH], helix C[E(6)Y], helix D [ExxH]) similar to those characteristic of two ferritin-like protein classes: erythrins and manganese catalases (*Andrews, 2010*).

Our homology-based tertiary structure modeling was performed using ModWeb (*Eswar et al., 2003*). The helix A/B and helix C/D loops (residues 57–84 and 109–152, respectively) were optimized with Rosetta3 kinematic loop closure in centroid mode (command line flag: -loops:remodel perturb_kic, loops were built from randomly selected cut-points)(*Leaver-Fay et al., 2011*; *Mandell et al., 2009*). The lowest energy structure was repacked in full atom fixed backbone design mode (with -ex1 -ex2aro flags), Thus, the CDS3 model spans residues 33–202 of the 290 total residues.

Crystal structures of three Mn catalases, from *Thermus thermophilus*, *Lactobacillus plantarum*, and *Anabaena PCC 7120* have been solved (*Antonyuk et al., 2000*; *Barynin et al., 2001*; *Bihani et al., 2013*). They are globular homohexamers and contain canonical ferritin four-helix bundles with notable features including a short (~20) residue N-terminal segment before Helix A; a quite long (~40–60 residue) cross-over loop between helices B and C (as a comparison the crossover in a pseudo-nitzschia diatom ferritin is 20 residues [*Marchetti et al., 2009*], and a long C-terminal tail (of 50–100 residues) that wraps around a neighboring monomer in the homohexamer complex. Notably, the CDS1-3 sequences contain both a predicted N-terminal segment (of 10 to 33 residues) before the assumed start of helix A and a C-terminal tail of ~70–100 residues. The predicted crossover loop in the algal sequences has a mid length (~40 residues) between those found in ferritins and the structurally characterized Mn catalases. In the known Mn catalase structures, the long crossover loops constitute much of the contact surface between chains in the homohexamer, and we speculate that the additional presumptive loop between helices A and B in the algal proteins may participate in similar inter-chain binding interactions.

## Morphological analysis

We observed cells with an Olympus BX53 microscope using a UPlanFL N 100x/1.30 Oil Ph 5 UIS two objective (Olympus Corporation, Tokyo, Japan) using brightfield display or fluorescence. An X-cite series 120 Q fluorescent illumination source (Lumen Dynamics, Ontario, Canada) was used to visualize Bodipy-505/515 stained cells. Bodipy 505/515 absorbs at 488 nm and emits at 567 nm. Lipid staining was performed as follows: 1 µl of 10 mg/mL Nile-red or Bodipy-505/515 and Triton X-100 (0.1%) was added to 100 µl of cells (concentration of appx. $1 \times 10^6$) and allowed to incubate at room temperature for one hour. Cell size analysis was performed using a Cellometer Auto M10 from Nexcelcom Bioscience (Lawrence, MA, USA). $2 \times 20$ µl from each liquid algal sample was analyzed for the live/dead count, total cell count, mean diameter (µm), viability (%), and live cell concentration.

## Bioreactor growth

*Chloroidium sp. UTEX 3007* was maintained as single colony isogenic monocultures repeatedly throughout our experiments. Algal cultures were grown under constant illumination (400 umol photons $m^{-2} s^{-1}$) at 25°C. Media used for algal culture were modified F/2 media (*Tamburic et al., 2014*) enriched with nitrate (10 mM $KNO_3$) and magnesium (2 mM $MgSO_4$). Saltwater medium was made using Ao Reef Salt (Ao Aqua Medic, Bissendorf, Germany) to 40 g/L unless indicated otherwise. Growth was measured using the bioreactor growth chamber Multi-Cultivator MC 1000 and MC 1000-OD by Photon Systems Instruments (Drasov, Czech Republic). For artificial, open pond, simulation, we grew algae in outdoor pond simulator bioreactors (PBR-101, Phenometerics Inc., East Lancing, MI, USA). The open pond simulators were approximately cylinder photobioreactors (PBRs) and were set up with a working volume of 400 ml and a light depth in the culture of 14.0 cm. The PBRs were injected with air enriched with 2.0–3.0% $CO_2$ at a flow rate of 0.20 L/min. We maintained temperature and pH at $25 \pm 1°C$ and $7.0 \pm 0.2°C$, respectively. Stirring was set at a constant rate of 200 RPM using a 28.6 mm stir bar. The cultures were grown under a light-dark cycle of 16:8 using a sinusoidal approximation of daily light with a peak intensity of 2000 µmol photons $m^{-2} s^{-1}$.

Cells were prepared as described above and per Phenometerics Inc.'s instructions included in the bioreactor setup guide. Briefly, cells were inoculated from solid agar media into liquid media for 24 hr in illuminated flasks on a shaker in the growth chambers described above. After inoculation of 10 mL of pre-prepared (concentration of about $5 \times 10^7$ cells/mL) into 70 mL of media in each bioreactor tube (80 mL total liquid volume in eight tubes/bioreactor setup), optical density (OD) readings were recorded at 680 nm every hour for 8–12 days. Unless otherwise noted, bioreactors were maintained at 25°C and were illuminated at 400 µmol photons $m^{-2}$ $s^{-1}$. Cell counting was performed using a Cellometer Auto M10 from Nexcelcom Bioscience (Lawrence, MA, USA). $2 \times 20$ ul from each liquid algal sample was analyzed for the live/dead count, total cell count, mean diameter (µm), viability (%), and live cell concentration. Cell concentrations for growth curves were calculated by constructing a standard curve correlating to absorbance at 680 nm, which is roughly proportional to chlorophyll a concentrations, with measured cell concentrations, R > 0.99.

## Flow cytometry

Cells were harvested at one and three weeks of growth in PBRs to obtain cells in mid-log and stationary phases, respectively. Staining was performed according to the microscopy procedure (above). Forward scatter (FSC) signal, assumed to be proportional to cell size or cell volume, and side scatter (SSC) signal, related to the complexity of the cell (*Ramsey et al., 2016*), were collected and plotted against fluorescence from BODIPY 505/515 in the FITC-A channel.

## Biolog/Omnilog assays

Phenotyping was done using standard Biolog assay plates and using the Omnilog instrument (Biolog Inc., Hayward, USA) as previously described (*Chaiboonchoe et al., 2014*). In total, 380 substrate utilization assays for carbon sources (PM01 and PM02), 95 substrate utilization assays for nitrogen sources (PM03), 59 nutrient utilization assays for phosphorus sources, and 35 nutrient utilization assays for sulfur sources (PM04), along with peptide nitrogen sources (PM06-08) were performed (Dataset 2-biolog). A defined tris-acetate-phosphate (TAP) medium (*Gorman and Levine, 1965*) containing 0.1% tetrazolium violet dye 'D' (Biolog, Hayward, CA, USA) was used for the PM tests. The carbon, nitrogen, phosphorus, or sulfur component of the media was omitted from the defined medium when applied to the respective PM microplates that tested for each of those sources.

Cells were grown in new tris-minimal media to mid-log phase, then spun down at 2000 g for 10 min, and then re-suspended in fresh media to a final concentration of $1 \times 10^6$ cells/mL before inoculation into Biolog's 96-well plates. A 100 µL aliquot of cell-containing media was inoculated into each well before the plates were inserted into the Omnilog system. A final concentration of 400 µL/mL Timentin (GlaxoSmithKline, Brentford, UK) was used to inhibit bacterial growth in all plates. Bacterial contamination was monitored by streaking cells on yeast extract/peptone plates and performing gram stains before and after Biolog assays. All microplates were incubated at 30°C for up to 8 days, and the dye color change (in the form of absorbance) was read with the Omnilog system every 15 min. As the Omnilog instrument does not provide a source of continuous light during incubation, the algae are assumed to be carrying out heterotrophic respiration.

The Biolog Phenotype Microarray (PM) data analysis was conducted using an Omnilog Phenotype Microarray (OPM) software package that runs within the R software environment (*Vaas et al., 2013*, *2012*). The raw kinetic data were exported as csv files to the OPM package, and then the biological information was added as metadata (e.g. strain designation, growth media, temperature, etc.).

## Gas chromatography-flame ionization detection (GC-FID)

For the extraction, 50 mg algal material (dry weight) was used by adding 1 ml of 1 N HCl/methanol solution (Sigma-Aldrich, Darmstadt, Germany). An internal standard (100 µl of FA15:0, pentadecanoic acid) was added to each sample before incubation at 80°C in a water bath for 30 min. After cooling to room temperature, 1 ml of 0.9% NaCl and 1 ml of 100% hexane were added to each vial. Vials were shaken for 5 s and centrifuged for 4 min at 1000 rpm. The upper FAME-containing hexane phase was transferred to a new glass vial, where it was concentrated under a stream of $N_2$. Finally, FAMEs were dissolved in hexane and filled into GC glass vials. The details of the GC-FID method are as follows: injector temperature of 250°C; helium carrier gas; head pressure 25 cm/s (11.8 psi); GC column, J and W DB23 (Agilent, Santa Clara, CA, USA), 30 m 9 0.25 mm 9 0.25 lm; detector

temperature 250°C; detector gas H₂40 ml/min, air 450 ml/min, He make-up gas 30 ml/min. FAME peaks were identified by comparing their retention time and equivalent chain length on standard FAME. We used standards of fatty acid methyl esters (FAME) from Supelco 37 Component FAME Mix (CRM47885, SIGMA-ALDRICH, Darmstadt, Germany) and Supelco PUFA No.3, from Menhaden Oil (47085 U SIGMA-ALDRICH, Darmstadt, Germany). Source data can be found in Dataset 2.

### Gas chromatography-mass spectrometry (GC-MS)

Extraction and analysis by gas chromatography coupled with mass spectrometry was performed using the same equipment set up and exact same protocol as described in *Lisec et al. (2006)*. Briefly, frozen ground material was homogenized in 300 µL of methanol at 70°C for 15 min and 200 µL of chloroform followed by 300 µL of water were added. The polar fraction was dried under vacuum, and the residue was derivatized for 120 min at 37°C (in 40 µl of 20 mg ml-1 methoxyamine hydrochloride in pyridine) followed by a 30 min treatment at 37°C with 70 µl of MSTFA. An autosampler Gerstel Multi-Purpose system (Gerstel GmbH and Co.KG, Mülheim an der Ruhr, Germany) was used to inject the samples to a chromatograph coupled to a time-of-flight mass spectrometer (GC-MS) system (Leco Pegasus HT TOF-MS (LECO Corporation, St. Joseph, MI, USA)). Helium was used as carrier gas at a constant flow rate of 2 ml/s and gas chromatography was performed on a 30 m DB-35 column. The injection temperature was 230°C and the transfer line and ion source were set to 250°C. The initial temperature of the oven (85°C) increased at a rate of 15 °C/min up to a final temperature of 360°C. After a solvent delay of 180 s mass spectra were recorded at 20 scans s-1 with m/z 70–600 scanning range. Chromatograms and mass spectra were evaluated by using Chroma TOF 4.5 (Leco) and TagFinder 4.2 software. Source data can be found in Dataset 2.

### Qualitative ultra high-performance liquid chromatography coupled with quadrupole time-of-flight mass spectrometry (UHPLC/MS-QToF)

For the extraction, cultured algae was scraped from agar plates and placed into 5 mL methanol and vortexed. Algal-methanol solutions were microwaved on high power (1150 watts, 2450 Mhz) in a Samsung ME732K microwave with a triple distribution system (Samsung, Seoul, South Korea). Solutions were microwaved until boiling five times and then vortexed. Extracts were filtered with a 2 µm filter (Millipore (Merck Millipore, Billeric, MA, USA) and maintained in the dark at 4°C.

The HPLC/MS method was developed in-house and is based on a comprehensive shotgun lipidomic technique from the Castro-Perez laboratory (*Castro-Perez et al., 2010*). Four µl of the extract from each sample was injected into a reverse-phase C18 column heated to 50°C. A quaternary pump maintained a 300 µl/min flow rate of the solvent composite over the sample. The starting solvent was 36% water with 30 mM ammonium formate, 36% acetonitrile, and 28% isopropanol which approached 90% isopropanol and 10% acetonitrile in a semi-linear gradient over 18 min. The end stream was diverted to a quadrupole time-of-flight mass spectrometer in positive mode with accurate mass profiling enabled that was tuned within 1 hr before the experiments (reference compounds were 121.050873 and 922.009798 m/z) (Agilent LCMS QToF 6538 (Agilent, Santa Clara, CA, USA)). The data were processed using Agilent's software and XCMS with METLIN(*Sana et al., 2008*; *Smith et al., 2005*). Source data can be found in Dataset 2.

## Acknowledgements

Financial support for this work was provided by New York University Abu Dhabi (NYUAD) Institute Grant (G1205-1205i, −1205h, −1205e), and NYUAD Faculty Research Funds (AD060). We thank the NYUAD High-Performance Computing and Core Technology Platform for the support. We thank Marc Arnoux and Mehar Sultana for carrying out high throughput sequencing at the NYUAD Sequencing Core and NYUAD Bioinformatics Core for assisting in pipeline development.

## Additional information

### Funding

| Funder | Grant reference number | Author |
|---|---|---|
| NYUAD Institute | Grant (G1205-1205i) | Kourosh Salehi-Ashtiani<br>David R Nelson<br>Basel Khraiwesh |
| NYUAD Faculty Research Funds | AD060 | Kourosh Salehi-Ashtiani<br>Weiqi Fu<br>Ashish Jaiswal<br>Amphun Chaiboonchoe |
| NYUAD Institute | Grant (G1205-1205h) | Khaled M Hazzouri |
| NYUAD Institute | Grant (G1205-1205e) | Glenn L Butterfoss<br>Kristin C Gunsalus |
| NYUAD Institute | Grant (G1205-1205a) | Nizar Drou<br>Jillian D Rowe |

The funders had no role in study design, data collection and interpretation, or the decision to submit the work for publication.

### Author contributions

DRN, Conceptualization, Data curation, Software, Formal analysis, Validation, Investigation, Methodology, Writing—original draft, Project administration; BK, Conceptualization, Supervision, Methodology, Writing—review and editing; WF, Conceptualization, Formal analysis, Writing—review and editing; SA, Formal analysis, Methodology, Writing—original draft, Writing—review and editing; AJ, Conceptualization, Software, Formal analysis, Visualization, Methodology, Writing—original draft, Writing—review and editing; AC, Data curation, Software, Formal analysis, Visualization, Methodology, Writing—review and editing; KMH, Conceptualization, Software, Formal analysis, Investigation, Methodology, Writing—review and editing; MJO'C, Conceptualization, Resources, Data curation, Software, Formal analysis, Methodology; GLB, Conceptualization, Data curation, Software, Formal analysis, Investigation, Methodology, Writing—original draft, Writing—review and editing; ND, Conceptualization, Data curation, Software, Formal analysis, Investigation, Methodology, Writing—review and editing; JDR, Conceptualization, Data curation, Software, Methodology; JH, Conceptualization, Resources, Supervision, Investigation, Methodology, Project administration; ARF, Conceptualization, Resources, Formal analysis, Supervision, Validation, Investigation, Methodology, Project administration, Writing—review and editing; KCG, Resources, Supervision, Funding acquisition, Investigation, Methodology, Project administration, Writing—review and editing; KS-A, Conceptualization, Resources, Supervision, Funding acquisition, Investigation, Visualization, Methodology, Writing—original draft, Project administration, Writing—review and editing

### Author ORCIDs

David R Nelson, http://orcid.org/0000-0001-8868-5734
Basel Khraiwesh, http://orcid.org/0000-0001-7026-1263
Weiqi Fu, http://orcid.org/0000-0002-7368-383X
Saleh Alseekh, http://orcid.org/0000-0003-2067-5235
Ashish Jaiswal, http://orcid.org/0000-0002-6193-1824
Amphun Chaiboonchoe, http://orcid.org/0000-0002-2009-0806
Jamil Harb, http://orcid.org/0000-0001-6334-3746
Kourosh Salehi-Ashtiani, http://orcid.org/0000-0002-6521-5243

## Additional files

### Major datasets

The following dataset was generated:

**Database, license,**

| Author(s) | Year | Dataset title | Dataset URL | and accessibility information |
|---|---|---|---|---|
| Nelson DR, Khraiwesh B, Fu W, Alseekh S, Jaiswal A, Chaiboonchoe A, Hazzouri KM, O'Connor MJ, Butterfoss GL, Drou N, Rowe JD, Harb J, Fernie AR, Gunsalus KC, Salehi-Ashtiani K | 2017 | Data from: The genome and phenome of the green alga Chloroidium sp. UTEX 3007 reveal adaptive traits for desert acclimatization | https://dx.doi.org/10.5061/dryad.k83g4 | Available at Dryad Digital Repository under a CC0 Public Domain Dedication |

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
