## [Decision Letter]

Thank you for submitting your article "The genome and phenome of the green alga *Chloroidium sp. UTEX 3077* reveal adaptive traits for desert acclimatization" for consideration by *eLife*. Your article has been reviewed by two peer reviewers, and the evaluation has been overseen by a Reviewing Editor and Ian Baldwin as the Senior Editor.

The reviewers have discussed the reviews with one another and the Reviewing Editor has drafted this decision to help you prepare a revised submission.

Summary:

This manuscript describes the genome and phenome of the green algae *Chloroidium sp. UTEX 3007*. This species is interesting as it appears well adapted to a spectrum of environmental conditions, especially salt concentrations. From a biofuels perspective the alga is interesting because under certain growth conditions it can reach high concentrations of the fatty acid palmitic acid. Palmitic acid is suggested to benefit the alga under desert conditions. Overall, the manuscript is well written, but there are substantial concerns with the quality of the figures. One of the reviewers also expressed major concerns with methodological flaws.

Major revisions:

1) Methodological problem: The authors used "codon volatility" method to detect positive (Darwinian) selection acting on some genes. Although, the method sounds very attractive because it supposedly detects positive selection using a single sequence, it has been criticized on many occasions as being unreliable and inconclusive. Here are some papers evaluating robustness of the codon volatility method: Mol Biol Evol. 2005 Mar;22(3):542-6; Mol Biol Evol. 2005 Apr;22(4):807-9; Mol Biol Evol. 2005 Mar;22(3):496-500; Genetics. 2005 Jan;169(1):495-501. In my opinion the whole section on gene adaptation has little value and should be removed unless solid analysis using well established methods will be performed.

2) In general, figures lack sufficient description and the reader is left guessing what a given graph is actually supposed to represent. Some essential information is lacking in figures, e.g. units are not displayed on the axes, fonts used are too small. This lack of attention to the quality of figures left both reviewers quite unhappy. Following are some specific problems with the figures; however, this list is not comprehensive and the authors should very carefully review and revise their figures.

a) Figure 3 and Figure 4 are impossible to read, even the x and y labels.

b) Figure 6 presents two data sets: full and selected but there is no explanation as to the difference between these two sets.

3) Evidence of horizontal gene transfer in the presented data, as claimed in the subsection “Manganese catalase-like genes”, is not convincing. A comprehensive phylogenetic analysis would be required to demonstrate HGT.

4) Terminology problem. Homology is a binary term defined as features, including genes and proteins that share common ancestry. As such two proteins are either homologous or not. In particular, there's no "strong homology" (Figure 5 legend) but "strong similarity."

5) At the beginning of the genomic sections – –subsection “Genomic analysis”, – the authors could indicate how many contigs were obtained, how many chromosomes the algal might have, the G+C content of the genome and the percentage of the genome that is predicted to be protein encoding.

6) Supplementary materials mentioned in the second paragraph of the subsection “Genome assembly and annotation”, are not accessible.

---

## [Author Response]

*Major revisions:*

*1) Methodological problem: The authors used "codon volatility" method to detect positive (Darwinian) selection acting on some genes. Although, the method sounds very attractive because it supposedly detects positive selection using a single sequence, it has been criticized on many occasions as being unreliable and inconclusive. Here are some papers evaluating robustness of the codon volatility method: Mol Biol Evol. 2005 Mar;22(3):542-6; Mol Biol Evol. 2005 Apr;22(4):807-9; Mol Biol Evol. 2005 Mar;22(3):496-500; Genetics. 2005 Jan;169(1):495-501. In my opinion the whole section on gene adaptation has little value and should be removed unless solid analysis using well established methods will be performed.*

We thank the reviewer for this comment, and due to the limited availability of well-curated, closely related genomes, we were unable to do conventional analyses to support the codon volatility analysis, for that we have removed the codon volatility section and replaced it with a new section accompanied by a new figure (Figure 6 in the revised manuscript) that describes unique protein families in *Chloroidium sp. UTEX 3077* and how they relate to the phenotypes we observed. The new sections provide evidence of unique genes and protein families present in this alga that have potential roles in desert adaptation may be important for the phenotypes we documented. Well-curated genomes from 5 other microalgae, each representing a separate clade of green algae, were downloaded from NCBI and Phytozome database and used for the comparative analysis. Their respective citations were added to the manuscript in the bibliography and corresponding text locations, and we did structural and functional annotation of these genomes using the same methods (and parameters) that we did for *Chloroidium sp. UTEX 3077* in order to avoid obtaining false positives and negatives due to different annotation methodologies. The resultant protein family (Pfam) assignments can be viewed interactively in the new Figure 6 by uploading Dataset 3 (Dataset_3.ivenn) to www.interactivenn.net. Unique Pfams relevant to our phenotype studies can be found in Table 1. In-text additions can be found in the Results subsection "Genome-scale metabolic network reconstruction and oil accumulation pathways” and in the Discussion subsection “Genomics of Chloroidium”.

Briefly, we contrasted protein families from the different algae and highlight a suite of 247 genes that are unique to *Chloroidium sp. UTEX 3077* and may provide molecular basis for its successful habitation of the region. We describe subsets of these genes that are relevant to the carbon assimilation and euryhaline phenotypes we documented in the metabolomics studies. Selected Pfam domains from these collections are highlighted in a new Table 1 that replaces the previous Table 1.

*2) In general, figures lack sufficient description and the reader is left guessing what a given graph is actually supposed to represent. Some essential information is lacking in figures, e.g. units are not displayed on the axes, fonts used are too small. This lack of attention to the quality of figures left both reviewers quite unhappy. Following are some specific problems with the figures; however, this list is not comprehensive and the authors should very carefully review and revise their figures.*

*a) Figure 3 and Figure 4 are impossible to read, even the x and y labels.*

We have revised the legends to provide more information for reader interpretation. Additional information is given on methodology, key findings, and links to the raw datasets and interactive plots. Likewise, the figures have been edited for clarity. The edits include rearrangement of panels among figures and the addition of new panels. Larger fonts, together with higher resolution figures, help to assist figure interpretation.

All source image files and their source data accompany the re-submission. We uploaded the 2 major datasets (Genomic and Phenomic datasets; Dataset 1 and Dataset 2, respectively; first mentioned and referenced in the second paragraph of the Introduction) at Dryad Digital Repository (64).

Specific figure changes:

1) Figure 1 now contains three panels: (A) a map of the UAE showing locations where *Chloroidium sp. UTEX 3077* strains were isolated from, (B) an Abu Dhabi map showing locations of isolation solely within Abu Dhabi, and (C) eight sub-panels illustrating morphology and lipid accumulation phenotypes of the species through phase contrast and fluorescence microscopy.

2) The original Figure 2 has been renamed to ‘Figure 3’. The new Figure 2 now contains panels with data previously associated with Figure 1, including (A) cell size distribution data, (B) photobioreactor growth data, and (C) flow cytometry data. Figure 2) now summarizes flow cytometry data using a “box-and-whiskers” plot to represent population data as opposed to the previously used histogram.

3) Figure 3 was renamed Figure 4. The new Figure 3, previously named ‘Figure 2’, was edited for clarity and homogeneity. The histogram for *Chloroidium sp. UTEX 3077* in panel 3B was corrected to reflect true values obtained in the GC-FID experiment (a calculation error in the original manuscript caused the lipids by dry weight to equal ~65%, where the true value was 78.1%, and the percentage of palmitic acid in the final culture was corrected from 80% to 41.8% (nearly the same as palm oil (43%)). Additionally, the comparison between *Chloroidium sp. UTEX 3077* and *Chlamydomonas reinhardtii* HPLC-MS results was expanded upon in the text (subsection “Intracellular metabolite profiling”, second paragraph).

4) Figure 4 was removed, and its data is now disseminated in the text. The new Figure 4, containing the data from the previous Figure 3, was edited for clarity by removing smaller text and highlighting compounds of interest. The complete listings of compounds are still available in the linked datasets.

5) Domain annotations on Figure 5 were abbreviated and described in the (expanded) legend. Panel (B) was removed as we decided that it did not add significantly to the manuscript.

6) Figure 6, previously describing the now-defunct codon volatility analysis, was replaced with a Venn diagram comparing common and unique PFAM domains in *Chloroidium sp. UTEX 3077* and 5 other species of algae (as described in the response to revision 1).

7) The font on Figure 7 has been enlarged for clarity.

*b) Figure 6 presents two data sets: full and selected but there is no explanation as to the difference between these two sets.*

This figure has been removed along with the codon volatility analysis.

*3) Evidence of horizontal gene transfer in the presented data, as claimed in the subsection “Manganese catalase-like genes”, is not convincing. A comprehensive phylogenetic analysis would be required to demonstrate HGT.*

The reference to horizontal gene transfer origin for this gene has been removed in accordance with the reviewer’s suggestion. We attempted to reconstruct phylogenies based on nucleotide data in available databases but did not obtain high bootstrap support for any one model. We used MUSCLE (100 iterations) to align genes from the 50 top BLASTN hits with the *Chloroidium sp. UTEX 3077* genes and performed maximum likelihood phylogenies with 100 bootstrap replicates. Several important branches in the model had bootstrap support lower than 40; therefore, one cannot confidently draw conclusions on the HGT origin of these genes with the available data.

*4) Terminology problem. Homology is a binary term defined as features, including genes and proteins that share common ancestry. As such two proteins are either homologous or not. In particular, there's no "strong homology" (Figure 5 legend) but "strong similarity."*

Quantitative adjectives for ‘homology’ redirected towards ‘similarity’.

*5) At the beginning of the genomic sections – –subsection “Genomic analysis”, – the authors could indicate how many contigs were obtained, how many chromosomes the algal might have, the G+C content of the genome and the percentage of the genome that is predicted to be protein encoding.*

Detailed statistics of the assembly, including how many contigs were obtained, the G+C content of the genome and the percentage of the genome that is predicted to be protein encoding, an estimate of chromosome count (n=16) were added to the beginning of the genomic Results subsection “Genomic analysis”.

*6) Supplementary materials mentioned in the second paragraph of the subsection “Genome assembly and annotation”, are not accessible.*

Accompanying datasets are available as described above in the response to the suggested revision #2. The Dryad datasets can be accessed at http://dx.doi.org/10.5061/dryad.k83g4, and the Plot.ly datasets at https://plot.ly/~davidroynelsonnyuad/1302/?share_key=NiqPEo2iFkCV7P0SBs8z65.